behaviour

ultrasonic vocalizations, social behaviour, affiliation, social cognition and memory

**Author for correspondence:**
Kouta Kanno
e-mail: canno@leh.kagoshima-u.ac.jp

†These authors contributed equally to this work.

# Sex differences in vocalizations to familiar or unfamiliar females in mice

Eri Sasaki†, Yuiri Tomita† and Kouta Kanno

Laboratory of Neuroscience, Course of Psychology, Department of Humanities, Faculty of Law, Economics and Humanities, Kagoshima University, Kagoshima City, Kagoshima, Japan

  KK, 0000-0003-0061-6158

Mice, both wild and laboratory strains, emit ultrasound to communicate. The sex differences between male to female (male–female) and female to female (female–female) ultrasonic vocalizations (USVs) have been discussed for decades. In the present study, we compared the number of USVs emitted to familiar and unfamiliar females by both males (male–female USVs) and females (female–female USVs). We found that females vocalized more to unfamiliar than to familiar females. By contrast, males exhibited more USVs to familiar partners. This sexually dimorphic behaviour suggests that mice change their vocal behaviour in response to the social context, and their perception of the context is based on social cognition and memory. In addition, because males vocalized more to familiar females, USVs appear to be not only a response to novel objects or individuals, but also a social response.

## 1. Introduction

Social animals use signals of specific modalities to communicate with other individuals. Among such signals, vocal communication is widely observed in animals and is superior in several aspects [1]. In mice, ultrasonic vocalizations (USVs) are used for communication, and they can be widely observed in both wild and laboratory strains [2]. There are two types of USVs commonly known in mice: pupUSVs and courtship vocalizations [3]. Mice pups emit ultrasounds (pupUSVs), mainly when they are outside their nest in the first several days after birth. A state of arousal due to the perception of cold, perception of unusual tactile stimulation or loss of social contact trigger the emission of pupUSVs, which enhance the mother's approach to the sound source [4]. The mother's attention and movement lead to the expression of maternal behaviour. Contrarily, adult male mice emit ultrasounds in the presence of adult females or their urine [5]; these are known as courtship vocalizations or courtship songs [6]. Activational effects of sex hormones on male vocalizations have been reported [7,8]. Furthermore, the number of male USVs is enhanced by sociosexual

experience with females [9], and usage of syllable types (call patterns) is modulated according to the progressive phase of sexual behaviours [10]. Thus, USVs from males to females are thought to be male-specific precopulatory behaviours [9,11]. In addition, several studies have revealed that male USVs in turn affect female behaviours. Females of Swiss-Webster mice spent more time with intact vocalizing males than with devocalized males in the preference test. This preference could be mimicked by playback sounds and was not observed in ovariectomized females [12]. Playback of male vocalizations in C56BL/6 (B6) strain also elicited approach behaviours in females [13], and playback of male USVs activated kisspeptin neurons, which play important roles in reproduction, as indicated by phosphorylated cyclic AMP response element binding protein (pCREB) immunoreactivity [14]. Moreover, females of B6 and BALB/c strains preferentially approach the playback vocalizations of the opposite strain [15], suggesting that females can discriminate acoustic features of vocalization to a certain extent.

In this regard, vocalizations in mice involved in these two contexts have been well investigated since they were first reported approximately 50 years ago [16–18]. Adult females were initially not thought to vocalize either to males or females [19]. However, using several strains, Maggio & Whitney [20] meticulously revealed that adult females emit USVs to females (female–female), whereas they rarely emit to males (female–male) and that the number of female–female USVs is comparable to male–female courtship USVs. Thereafter, D'Amato & Moles [21] studied the female–female USVs in albino NMRI female mice and found that females vocalize more to unfamiliar females than to familiar ones. This finding suggests that female–female USVs reflect social cognition and memory. In fact, administration of a classical cholinergic antagonist (scopolamine) disrupted this social memory [21]. Although female–female USVs are known to be related to this social memory, the functions of these USVs are poorly understood.

In the study by D'Amato & Moles [21], familiar individuals were those that were previously presented only several minutes in advance (while simultaneously recording ultrasound). Therefore, we decided to use female individuals that were reared in the same cage after weaning as familiar individuals. In other words, we hypothesized that female–female USVs reflect social affiliation if females emit USVs more to familiar females than to unfamiliar ones under such conditions. If this hypothesis is correct, the results of the study by D'Amato and Moles and our expected results could be explained as follows: females express high levels of USVs to establish an affiliative relationship with a stranger of the same sex first; and, at the same time, females also emit more USVs to familiar social members. In the study by D'Amato and Moles, the authors applied 15 min, 30 min, 60 min and 24 h time intervals after female subjects encountered females that were used as familiar individuals. Subject females emitted more USVs to unfamiliar than to familiar females under three different time intervals (15, 30 and 60 min), while no difference was observed after the 24 h interval. The effect of the cholinergic antagonist mentioned above was investigated at 30 min intervals because it represented the best interval for drug absorption. Thus, in the present study, a 30 min interval that seemed suitable for retention with versatile use and a 24 h interval that seemed severe for mice retention, was applied to B6 mice.

In the present study, we also compared female–female vocalizations and male–female vocalizations using the B6 strain, which is currently the most commonly used laboratory strain. Social cognition and memory in male mice have not yet been measured in this context; however, it is important to measure the sociality of both sexes using the same methods for future biomedical studies. Recently, vocal communication in mice has received considerable academic attention because of its power and utility in the investigation of molecular and neural mechanisms for social behaviours and their deficits, especially focusing on developmental disorders such as autism spectrum disorders [3,22,23]. For instance, in our previous study, mice carrying a mutation in the autism susceptibility candidate 2 (AUTS2) gene show an increase in excitatory synaptic inputs in the forebrain, as well as a reduction in their social interaction and vocalizations toward unfamiliar individuals [24]. Therefore, using both male and female mice, we aimed to observe alterations in USVs depending on different social contexts such as familiarity with social partners.

# 2. Material and methods

## 2.1. Animals

We used an inbred strain of B6 (C57BL/6 J) mice. Mice were purchased from Japan SLC (Hamamatsu, Japan). Subject animals were used when they were 8–10 weeks old for the recording tests. All mice, after being purchased were reared in the same breeding room. The food (5L37 Rodent LabDiet EQ,

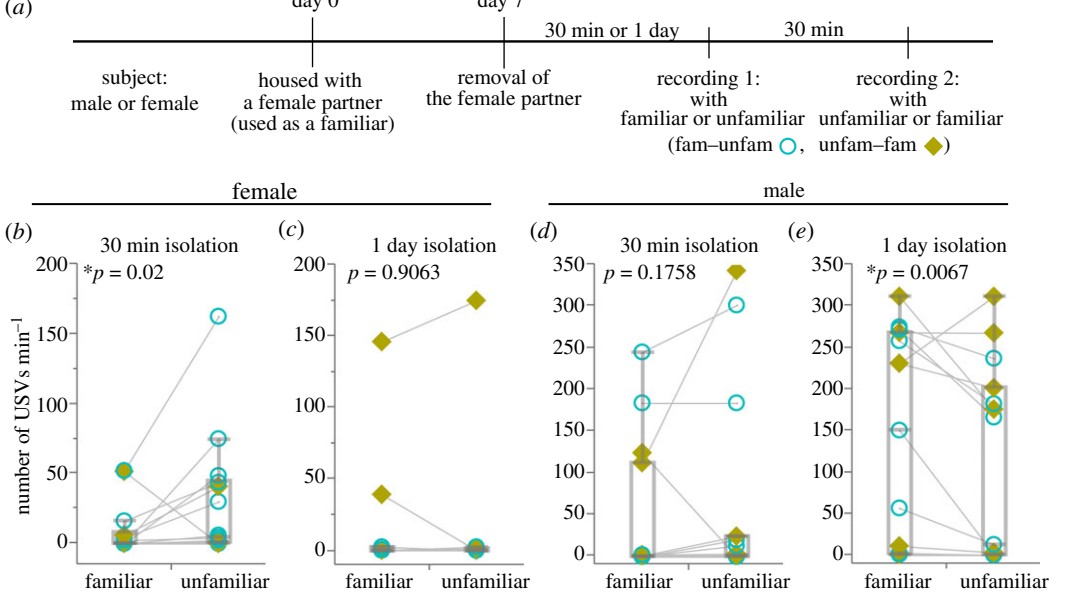

**Figure 1.** Male and female USVs to familiar or unfamiliar females. (*a*) Schematic diagram of the experimental procedure. (*b*) Female USVs to the familiar or unfamiliar females under the condition of 30 min isolation before recording 1. (*c*) Female USVs to the familiar or unfamiliar females under the condition of 1 day isolation before recording 1. (*d*) Male USVs to the familiar or unfamiliar females under the condition of 30 min isolation before recording 1. (*e*) Male USVs to familiar or unfamiliar females under the condition of 1 day isolation before recording 1. *$p < 0.05$, Wilcoxon signed-rank test; (*b*), $n = 11$; (*c*), $n = 14$, (*d* and *e*), $n = 15$. Counter-balanced order of experimental contents for recording 1 and 2 in each individual is indicated as follows: blue open circle, recording 1 with familiar and recording 2 with unfamiliar female (fam–unfam); lime yellow closed square, recording 1 with unfamiliar and recording 2 with familiar female (unfam–fam).

PMI Nutrition International, MO, USA) and bedding (soft woodchip, Japan SLC, Hamamatsu, Japan) were sold by the same supplier, Japan SLC.

For recordings of male–female USVs, a male and an age-matched unfamiliar female mouse were cohoused in a standard cage ($182 \times 260 \times 128$ mm, CREA Japan) immediately following arrival from the supplier until experiments began. Pups were delivered in four of the 29 male–female pairs. Owing to such small $N$ values, observation of delivery was not taken into account for the analysis.

For recordings of female–female USVs, females that were bred together in the same cage after weaning were purchased, and two individuals were housed in each cage immediately after arrival from the supplier until experiments began. One was used as the subject, and the other was used as the familiar partner (figure 1*a*).

For both recordings, age-matched, unfamiliar adult females were also purchased, and three to four animals were housed in the same cage accordingly. These animals were used repeatedly as an unfamiliar intruder for recording (see below and figure 1*a*).

Food and water were supplied ad libitum, and the animals were kept under a standard 12 h : 12 h light–dark cycle. All experiments were conducted in the light phase. The environment was maintained at a constant temperature (22–25°C) and humidity (50% ± 5%). All experimental procedures were approved by the Institutional Animal Use Committee of Kagoshima University (#L18004 and #L19003).

## 2.2. Experimental design and general procedures

The subject male was cohoused with an unfamiliar female, and the subject female was cohoused with a female that was bred together with the subject female in the same cage after weaning, as described above. These male–female and female–female mice were cohoused for 7 days without exchange of beddings. On the 7th day (Day 7 shown in figure 1*a*) after the pairing of male–female and female–female began (Day 0 shown in figure 1*a*), the female partner was removed. Recordings of USVs were conducted twice (recording 1 and 2) for all subjects, and two experimental conditions were tested in both male–female and female–female contexts. Some subjects were used for recording 1, conducted 30 min after the removal of the partner, and the other subjects were used 1 day (24–25 h) after the removal. Recording 2

was conducted 30 min after recording 1 for all subjects. USVs were induced by introducing a female mouse. The subjects were exposed to the familiar partner and an unfamiliar female once each, and the order of encountering the familiar or unfamiliar partner was counter-balanced between recordings 1 and 2.

## 2.3. Settings and procedures for ultrasound recordings

The microphone was hung 16 cm above the floor in a sound-attenuating chamber. Immediately prior to recording, the home cage of the subjects was moved into the chamber, where a red dim light was placed. The ultrasound recording was performed for 100 s after the intruder was introduced into the home cage.

According to previous studies [10,25,26], we conducted recordings without the devocalization of female encounters under natural conditions. Previous experiments suggested that most vocalizations are produced by the resident subjects and that the vocal contribution of female encounters is very limited in the male–female and female–female contexts [27–29].

## 2.4. Ultrasound analysis

Most of the settings for the USV recordings followed those of our previous study [9], with some modifications. USVs were recorded using a CM16/CMPA condenser microphone (Avisoft Bioacoustics, Berlin, Germany), ultrahigh-speed ADDA converter BSA768AD-KUKK1710 and its software SpectoLibellus2D (Katou Acoustics Consultant Office, Kanagawa, Japan) with a sampling rate of 384 kHz (to measure 20–192 kHz).

Recorded sounds were saved on a computer as wav files using SpectoLibellus2D. These sound files were analysed with the GUI-based software USVSEG (implemented as Matlab scripts) recently developed by us [30]. Continuous sound signals with frequencies ranging from 40 to 160 kHz and durations ranging from 3 to 300 ms were analysed and detected as syllables. Each syllable was segmented and saved as a JPEG image file, and the noise data (false positives) were manually excluded. Finally, the number of vocalizations (syllables) was quantified accordingly.

## 2.5. Statistics

Statistical analyses were performed using JMP 14 software (SAS Institute, Cary, NC, USA). To compare the two conditions within an individual, Wilcoxon signed-rank tests were applied (figure 1). To compare the two conditions between groups, Mann–Whitney $U$-tests were applied. Kruskal–Wallis tests were used for comparison among groups, followed by *post hoc* Steel–Dwass tests (see figures 2 and 3). Values were reported as dot plots with boxplots. Differences with $p < 0.05$ were considered significant.

# 3. Results

## 3.1. Female–female vocalization

Under the 30 min isolation condition, the number of USVs emitted from subject females to unfamiliar females was significantly higher than that from familiar partners (figure 1*b*; Wilcoxon signed-rank test, $W_{14,14} = 39.0$, $p = 0.02$). By contrast, no significant differences were detected between the number of USVs emitted to the unfamiliar and familiar partner after 1 day of isolation (figure 1*c*; Wilcoxon signed-rank test, $W_{11,11} = 3.0$, $p = 0.9063$). Furthermore, using only the data obtained from recording 1 (halving N values), the effect of familiarity of encountering females as a between-factor was compared using the Mann–Whitney $U$-test, and no significant differences were observed under both 30 min ($p = 0.7286$) and 1 day ($p = 0.7511$) isolation conditions.

## 3.2. Male–female vocalization

Under the 30 min isolation condition, no significant differences were detected between the number of USVs emitted from males to unfamiliar and familiar partners (figure 1*d*; Wilcoxon signed-rank test, $W_{15,15} = 26.0$, $p = 0.1758$). Contrarily, the number of USVs emitted to the familiar partner was significantly higher than that of the unfamiliar female after 1 day of isolation (figure 1*e*; Wilcoxon signed-rank test, $W_{15,15} = -47.5$, $p = 0.0067$). Additionally, using only the data obtained from recording 1, the effect of familiarity of encountering females as a between-factor was compared using the

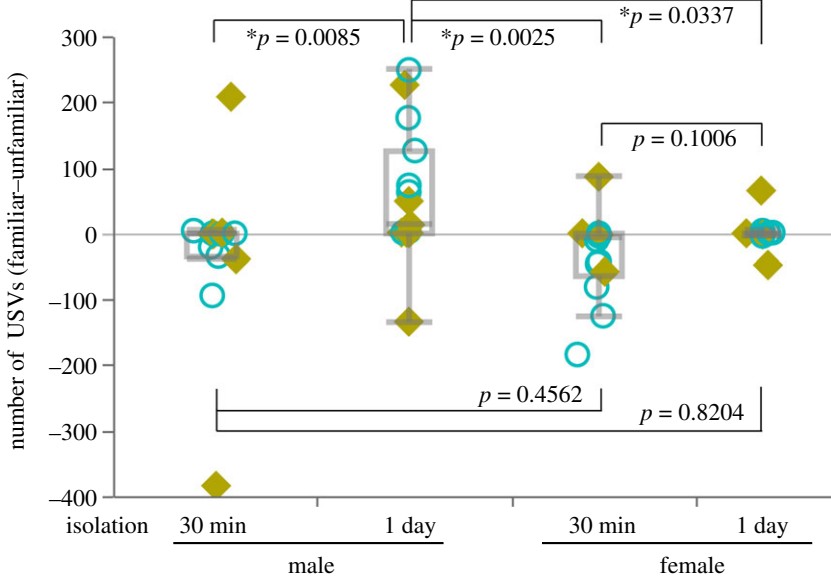

**Figure 2.** Sex differences in vocal response to familiar or unfamiliar females. A total comparison among groups is shown in figure 1. *$p < 0.05$, Steel–Dwass test after Kruskal–Wallis test; male 30 min and 1 day isolation, $n = 15$; female 30 min isolation, $n = 11$, female 1 day isolation, $n = 14$. The counter-balanced order of experimental contents for recording 1 and 2 in each individual is indicated in figure 1: Blue open circle, fam–unfam; lime yellow closed square, unfam–fam.

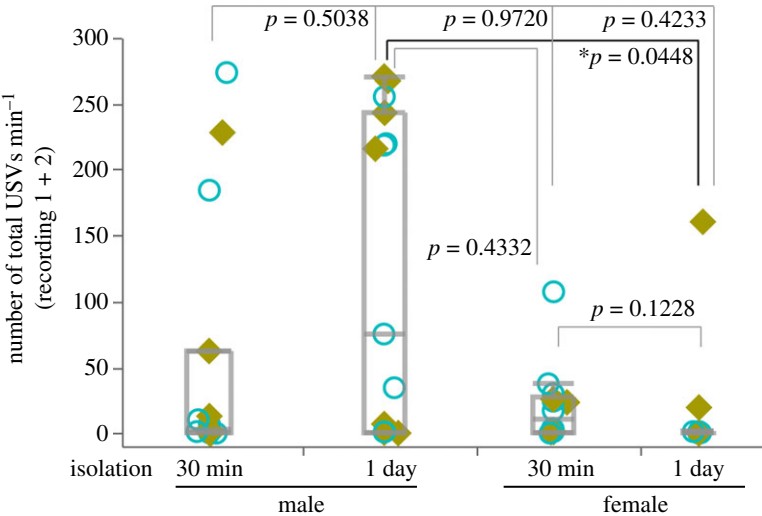

**Figure 3.** Comparison of the total number of USVs among groups. The total number of USVs (recording 1 + 2) per min among groups is shown. *$p < 0.05$, Steel–Dwass test after Kruskal–Wallis test; male 30 min and 1 day isolation, $n = 15$; female 30 min isolation, $n = 11$, female 1 day isolation, $n = 14$. The counter-balanced order of experimental contents for recording 1 and 2 in each individual is indicated as in figures 1 and 2: Blue open circle, fam–unfam; lime yellow closed square, unfam–fam.

Mann–Whitney $U$-test, and no significant differences were observed under both 30 min ($p = 0.9514$) and 1 day ($p = 0.9999$) isolation conditions.

## 3.3. Total comparison among groups

Differences between the number of USVs emitted to familiar and unfamiliar partners ([no. familiar]–[no. unfamiliar]) were calculated from the data shown in figure 1 and were compared among the four groups (figure 2). The Kruskal–Wallis test indicated significant differences between groups ($\chi^2 = 19.4288$, d.f. = 3, $p = 0.0002$) and the Steel–Dwass test (performed *post hoc*) showed that the value for males under the 1 day isolation condition was significantly higher than that of males or females under the 30 min isolation condition ($p = 0.0085$ and $p = 0.0025$, respectively), or females under the 1 day isolation

condition ($p = 0.0337$). The values for males under the 1 day isolation condition versus females under the 30 min ($p = 0.4562$) or 1 day ($p = 0.8204$) isolation conditions were comparable. Additionally, the values observed in females under the 30 min versus 1 day isolation conditions were not significantly different ($p = 0.1006$).

Additionally, the total number of USVs were compared in the two recording tests (recording 1 + 2). The number of USVs in the 200 s recording was calculated as USVs per minute in order to discuss the results with those of previous studies (figure 3). The Kruskal–Wallis test indicated significant differences between groups ($\chi^2 = 9.0411$, d.f. = 3, $p = 0.0287$) and the Steel–Dwass test (performed *post hoc*) showed those the value of males under the 1 day isolation condition was significantly higher than that of females under the 1 day isolation condition ($p = 0.0448$). No significant difference was observed in the other pairwise comparison (male 30 min versus male 1 day, $p = 0.5038$; male 30 min versus female 30 min, $p = 0.9720$; male 30 min versus female 1 day, $p = 0.4233$; male 1 day versus female 30 min, $p = 0.4233$; female 30 min versus female 1 day, $p = 0.1228$).

# 4. Discussion

## 4.1. Female–female vocalization

In this study, we used female subjects who spent more time with a female partner; the relationships between them are thought to be affiliative. Nevertheless, contrary to our expectation, the result of the previous study by D'Amato & Moles [21] was replicated in our relatively short isolation condition (figure 1*b*). We found that females vocalized more to an unfamiliar female than to familiar females after relatively short isolation intervals. It is unclear why females rarely vocalized under the relatively long isolation conditions, but no difference was found between the number of USVs to the unfamiliar and familiar females under such conditions, as was the case in the study by D'Amato and Moles. Thus, we believe, that B6 females vocalize more to unfamiliar females, after short isolation intervals.

## 4.2. Male–female vocalization

By contrast, it was surprising that males exhibited an opposite tendency toward females, after long isolation intervals. It has been reported that male courtship USVs are enhanced by sociosexual experience [9] and that vocal usage is altered according to the progressive phase of sexual behaviour [10]. Furthermore, castration reduces the USVs, while replacement of sex hormones restores the vocalizations [31]. In particular, the implantation of testosterone in both the ventral tegmental area and the medial preoptic area, which is known as a neural circuit for sexual motivation, effectively restores the USVs in castrated male mice [8]. These results indicate that male USVs directed towards females express sexual motivation as Nunez *et al.* [31] suggested that 'measures of male vocalizations provide an index of sexual motivation independent of male copulatory performance'. We expected males to show more vocalization to unfamiliar females because it has been reported that males exhibit more sexual motivation toward unfamiliar novel females (the Coolidge effect) [32,33]. However, contrary to our expectation, males made many vocalizations to the familiar partner. Even though different experimental conditions, such as length of cohousing, could possibly have led to different results, it appears that males exhibited more sexual motivation toward the familiar partner (cohoused for one week) under the present conditions.

## 4.3. Technical limitations and considerations for experimental design

It remains unclear why the optimal conditions for observation of differential vocal responses to the familiar or unfamiliar females differ between males and females. Additionally, there was a potential design flaw in the present paradigm. A major difference between the female–female pairs and the male–female pairs is that the female–female pairs were housed together since the time of weaning, whereas the male–female pairs were housed together only for 7 days. This makes interpretation of the sex differences difficult. Nevertheless, sex differences in such vocal responses are still considered important. For decades, the sex differences between male–female and female–female USVs have been discussed [19,20,28]. The amount of USVs between male–female and female–female is comparable [20]. In addition, the vocal repertoire between sexes has also been reported to be comparable [28]. By contrast, the sex differences shown in the present study indicate that mice change their vocal

behaviour in response to whom they are facing; their perception of the context is based on social cognition and memory. As observed in the female vocal response—this trend in response was still reproduced in the present study under different conditions from a previous study—mice usually exhibit attention to novel things or individuals. In some behavioural tests, such as habituation–dishabituation test, this property is being used accordingly [34,35]. However, because males vocalized more to familiar females after relatively long intervals, USVs are thought to be not only a response to novel things or individuals but also a social response. These findings with wild-type B6 mice provide a practical behavioural assay for the disease model mice, such as genetically modified mice with the B6 background, focusing on their social impairment. This is because if typical social behaviour can be described in the wild-type both in quality and quantity, then a valid assay for the detection of atypical characteristics as a deviation from *normal* in such a disease model can be established accordingly.

Another issue is that the paired experimental design seems problematic in combination with the fact that there was a 30 min delay between exposure to the familiar and unfamiliar female; the memory retention period was 30 min for the first exposure and 60 min for the second exposure. Therefore, although some would argue that between-comparison would have been better, there is yet another concern. There are large individual differences, at least in male–female USVs [9]. In fact, using only the data from recording 1, we compared the effect of familiarity of encountering females as a between-factor; however, no significant differences were observed among the groups. In such cases, a better way to observe the effects of the experiment would be to take repeated measurements while taking individual differences into account. Notably, the present study demonstrated such significant differences using a within-experimental design. In addition, since differential vocal response in subject females to familiar or unfamiliar ones was significantly observed with 30 and 60 min retention times, as mentioned in the Introduction [21], we speculate that our present paradigm of repeated measurements could detect such differences before the extinction of memory for social partners.

Furthermore, it is unclear why females rarely vocalized under the long isolation condition. Similarly, depending on the isolation conditions, the amount of USVs seems different in males. This might be explained by sexual refractory periods in males since they do not exhibit sexual behaviours (mounting and intromissions) after ejaculation. This interval between sexual behaviours is called a refractory period and USVs are not observed in this period (the first refractory period ranges from 30 to 60 min) [36]. In the present study, males vocalized more under the long isolation than under the short isolation condition, possibly because males were thought to be sexually satiated as a result of the 7 days of cohousing with a female before experiments. This relatively long isolation condition was thought to restore their sexual motivation and was tested because we expected this possibility. Nevertheless, it must be noted that the observed USVs in this study are low not only for females but also for males. The mean and median of USVs min$^{-1}$ in this study (figure 3) are as follows ([mean, median]): male 30 min isolation [52.3, 3]; male 1 day isolation [120.6, 75.3]; female 30 min isolation [19.7, 10.2]; female 1 day isolation [16.7, 0.3]. By contrast, the number of USVs min$^{-1}$ reported in other studies, including our previous study, is higher. For example, the mean number of approximately 250 min$^{-1}$ [9,24,37] or approximately 100–150 min$^{-1}$ [25,28] of male–female USVs, and approximately 150–200 min$^{-1}$ of female–female USVs have also been reported [25,28]. In these studies, several days (up to 7 days) of single housing was carried out in both male and female subjects. This is because sexual motivation is enhanced—and this motivation leads to increased USVs—for males, as mentioned above. In addition, Ey *et al*. [25] explained 'Tested females were isolated 3 days before the resident–intruder test to increase their motivation for social interactions'. Thus, in general, several days of isolation are apt to enhance USVs, even though long-term isolation cannot be used before recording to retain the memory of a social partner. Therefore, such studies focusing on social relations that are formed in the process of relatively long-term social interaction must be limited in their experimental paradigm. Even under such restrictions, we reported new findings in this study. For example, although the reason or mechanisms of isolation condition affecting the amount of USVs was not identified, significant differences observed under certain conditions were dependent on differences in social partners (familiar or unfamiliar).

## 4.4. Functional consideration of female–female USVs

Although such sex differences in USVs have been revealed, the past and present studies have yet to determine the biological significance and function of female–female USVs. Since Maggio & Whitney [20] hypothesized that female vocalization contributes to the formation of social hierarchy among females, this hypothesis has been referred to in some studies [21,28]. However, to our knowledge, this

hypothesis has never been investigated experimentally in mice. A recent study showed an important finding about sex differences in acoustic features, with more diversity (inter-individual difference) observed in the vocal repertoire in males than in females [38]. Nonetheless, the biological significance and function of female USVs remains unclear. Under such circumstances, we could speculate, at least a possible role based on social behaviours as already observed among females. It has been reported that female house mice display non-random preferences to other females, that is, a social partner choice. The partner choice in pairs with a preferred one leads to a higher probability of giving birth and establishing a cooperative relationship for communal nursing of offspring, resulting in higher reproductive success [39]. Female–female USVs may function in the process of such partner choice; however, further studies focusing on clarifying the roles of the USVs, including investigation of the associated neural mechanisms, are needed.

## 5. Conclusion

We found that females vocalized more to unfamiliar than to familiar females after a relatively short isolation interval in B6 strains, which could be similar to that observed in another strain. By contrast, males exhibited more USVs to familiar partners after a relatively long isolation interval. This sexually dimorphic behaviour suggests that mice change their vocal behaviour in response to the social context, and that their perception of the context is based on social cognition and memory. In addition, because males vocalized more to familiar females, USVs appear to be not only a response to novel things or individuals, but also a social response. The mechanisms and roles of these behaviours should be investigated further; in addition, these behaviours can be useful assays for social behaviour to compare with some transgenic strains, which can be applied to both males and females.

Ethics. All experimental procedures were approved by the Institutional Animal Use Committee of Kagoshima University (#L18004 and #L19003).

Data accessibility. The dataset supporting this article is available from the figshare Digital Repository: https://doi.org/10.6084/m9.figshare.12478646.v1.

Authors' contributions. K.K. contributed to the experimental design. E.S. and Y.T. conducted the experiments. E.S., Y.T. and K.K. contributed to the interpretation of data and writing of the manuscript.

Competing interests. We declare we have no competing interests.

Funding. This work was supported by JSPS KAKENHI grant nos. 18K13371 and 19H04912.

Acknowledgements. This work was supported by a Grant-in-Aid for Scientific Research on Innovation Areas 'Integrative Research toward Elucidation of Generative Brain Systems for Individuality' (19H04912) from MEXT to K.K. We thank various supports, including daily discussion, of this project. We would like to thank Editage (www.editage.com) for English language editing.

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
