## [Reviewer comments · Royal Society Open Science]

Review History

RSOS-201529.R0 (Original submission)

Review form: Reviewer 1

Is the manuscript scientifically sound in its present form?

Yes

Are the interpretations and conclusions justified by the results?

Yes

Is the language acceptable?

No

Do you have any ethical concerns with this paper?

No

Have you any concerns about statistical analyses in this paper?

Yes

Recommendation?

Accept with minor revision (please list in comments)

Comments to the Author(s)

This paper describes an experiment designed to test the vocal responses of mice to familiar and unfamiliar females using two different memory retention periods (30 min and one day). Females preferentially vocalized toward unfamiliar females after the 30 min retention period but not after the 1 day period. In contrast, males preferentially vocalized toward familiar females after the 1 day period but not after the 30 min period. The context of this experiment is limited in scope, and the authors are careful not to overstate the results. I reviewed a previously submitted version of this manuscript, and the authors have generally responded to my previous concerns adequately. My remaining concerns are itemized below:

1. The entire manuscript should be edited carefully by a native English speaker. Although many of the grammatical mistakes in the previous draft have been corrected, numerous new mistakes have been added.
2. The context of autism treatment is interesting (p. 4), but it would be helpful to explain how USV changed in mice with the manipulation of the AUT2 gene. Briefly mentioning this clinical context in the Discussion would also be useful.
3. Page 7 (lines 10-12) should be reworded as “. . . and 24-hour time intervals after female subjects encountered females were used as familiar individuals were applied.”
4. It remains vague regarding exactly what about this experiment was distinct from the results reported by D’Amat and Moles (2001). The main goals of this study, which are not simply a replication of past work, should be more clearly stated at the end of the Introduction.
5. On page 9 (line 5-7), it is stated that “Pups were delivered in 4 of 29 male-female pairs. Owing to small N values, observation of delivery was not taken into account for analysis”. This is a weak argument. 4 out of 29 is 14% of your sample. Were females separated from their pups to run the experiment? This would seem to be a serious confound, depending upon exactly when the pups were delivered. Perhaps eliminating these four pairs from the entire experiment would be reasonable?
6. On page 12 (line 16), it is stated that “In the present study, most of the data seemed o not be distributed normally . . .” This is a weak statement. On could run some analyses to determine if the samples are indeed non-normal (Shapiro-Wild statistic).
7. Page 15 (line 9), the phrase “of post hoc tests” can be deleted. More broadly, it would be useful to mention the non-significant p-values parenthetically throughout the Results section, but this decision should be made by the editor.
8. When describing the results in the Discussion, it should consistently be indicated that the significant difference in female vocalizations occurred after a short isolation interval and the effect occurred after a long isolation interval in males. For example, page 16 (line 6) should include “. . . unfamiliar females, after a short isolation interval.” This point should also be reiterated in the Conclusion section.
9. It is stated on page 17 (line 3-4), that “We expected males to show more vocalization to unfamiliar females”. The basis for this expectation should be briefly stated.
10. In general, the Discussion is quite wordy. Attention should be given to avoiding redundancy and eliminating irrelevant details. For example, the description of 30 min vs. 60 min delay could be stated much more concisely on pages 19-20.
11. On page 20, it is suggested that the differences in vocalization by males after 30 min vs. 1 day of isolation may have been due to the sexual refractory period. The typical length of the sexual refractory period in mice should be stated.

12. Page 20 (line 10), “saturated” should be “satiated”.
13. On page 18 (lines 7-9) the issue of differential response to novelty is brought up, but no sources are cited. It would be useful have some examples of how mice respond to novelty.

Review form: Reviewer 2

Is the manuscript scientifically sound in its present form?

Yes

Are the interpretations and conclusions justified by the results?

Yes

Is the language acceptable?

Yes

Do you have any ethical concerns with this paper?

No

Have you any concerns about statistical analyses in this paper?

No

Recommendation?

Accept with minor revision (please list in comments)

Comments to the Author(s)

This article has been modified in response to revisions requested on a prior submission to Biology Letters. Having read only this current version of the manuscript, I believe the authors have successfully answered most of the points previously raised. I have a few comments, raised below as I read through the manuscript:

Introduction

In the introduction, there is a bit of back-and-forth between ‘vocal communication’ and ‘autism’, which slightly impairs the flow of ideas. This could be improved by simple text rearrangements. For instance, the added section (lines 7-11) currently feels out of context because the authors suddenly refer to their work. Using more general terms could be more appropriate. For example, instead of ‘In addition, we previously investigated synapses and behaviors in the mutants of Autism susceptibility candidate 2 (AUTS2) gene, and we reported that excitatory synaptic inputs were increased in the forebrain and that social interaction and vocalizations were altered in the mice’, one could use ‘For instance, mice carrying the Autism susceptibility candidate 2 (AUTS2) gene show an increase in excitatory synaptic inputs in the forebrain, as well as alteration of their social interaction and vocalizations’. This is just a suggestion, which shows how the introduction could be framed better.

- Line 13: ‘Playback of male vocalizations in B6 strain’. This strain needs to be defined here as it is where it first appears.

- Currently, the end of the introduction reads ‘it is important to measure the sociality of both sexes using the same methods because mouse models and USVs have been recently investigated to clarify the basic mechanisms of developmental disorders [2–4]’. This section needs rewriting, as the suggested connection to developmental disorders is brought about without context.

Methods & Results

The methods and results are well described.

Discussion

- The authors did well in answering various points raised during the past revision rounds, gathered in a new section entitled '4.3 Technical limitations and considerations for experimental design'.

- In this section 4.3, the text reads 'For decades, the sex differences between male-female and female-female USVs have been discussed'. A reference would be useful here, in particular because the following sentence emphasizes the lack of sex difference in this context.

- I would suggest to replace 'mice change their vocal behavior in response to the context of whom they are dealing' by 'mice change their vocal behavior in response to whom they are facing' or similar.

- I would suggest to replace 'if typical social behavior can be observed in the wild-type both as a quality as well as a quantity' by 'if typical social behavior can be described in the wild-type both in quality as well as in quantity' or similar.

- Within section 4.3: I believe that the analysis conducted using only 'recording 1' is valuable (despite, unfortunate as it is, showing no significant findings) and should be included in the main document. This phase indeed represents the 'true' retention effect, in the sense that it comes right after the 30min or 1day isolation period.

- End of section 4.4: 'clarifying roles' should be 'clarifying the roles'

Conclusion

In the last sentence of the conclusion paragraph, I am not sure to understand the use of 'however'. Wouldn't 'in addition' be more appropriate for instance?

Decision letter (RSOS-201529.R0)

Dear Dr Kanno

On behalf of the Editors, we are pleased to inform you that your Manuscript RSOS-201529 "Sex differences in vocalizations to familiar or unfamiliar females in mice" has been accepted for publication in Royal Society Open Science subject to minor revision in accordance with the referees' reports. Please find the referees' comments along with any feedback from the Editors below my signature.

Please submit your revised manuscript and required files (see below) no later than 7 days from today's (ie 09-Nov-2020) date. Note: the ScholarOne system will 'lock' if submission of the

revision is attempted 7 or more days after the deadline. If you do not think you will be able to meet this deadline please contact the editorial office immediately.

Best regards,

on behalf of Dr Cynthia Downs (Associate Editor) and Kevin Padian (Subject Editor)
openscience@royalsociety.org

Associate Editor Comments to Author (Dr Cynthia Downs):

Two reviewers and I reviewed this manuscript. One reviewer had reviewed a previous version of this manuscript when it was submitted to the *Biology Letter*, and the other reviewer was new.

Both the reviewers and I appreciate the edits that the authors made in response to comments on the previous draft. In particular, the reviewers appreciated that the scope of inference was narrowed in this version of the manuscript. Most of the reviewers' comments are about clarifying phrasing or correcting errors in English grammar. I encourage the author to make the suggested changes.

Reviewer comments to Author:

Reviewer: 1
Comments to the Author(s)

This paper describes an experiment designed to test the vocal responses of mice to familiar and unfamiliar females using two different memory retention periods (30 min and one day). Females preferentially vocalized toward unfamiliar females after the 30 min retention period but not after the 1 day period. In contrast, males preferentially vocalized toward familiar females after the 1 day period but not after the 30 min period. The context of this experiment is limited in scope, and the authors are careful not to overstate the results. I reviewed a previously submitted version of this manuscript, and the authors have generally responded to my previous concerns adequately. My remaining concerns are itemized below:

1. The entire manuscript should be edited carefully by a native English speaker. Although many of the grammatical mistakes in the previous draft have been corrected, numerous new mistakes have been added.
2. The context of autism treatment is interesting (p. 4), but it would be helpful to explain how USV changed in mice with the manipulation of the *AUT2* gene. Briefly mentioning this clinical context in the Discussion would also be useful.

3. Page 7 (lines 10-12) should be reworded as “. . . and 24-hour time intervals after female subjects encountered females were used as familiar individuals were applied.”

4. It remains vague regarding exactly what about this experiment was distinct from the results reported by D'Amat and Moles (2001). The main goals of this study, which are not simply a replication of past work, should be more clearly stated at the end of the Introduction.

5. On page 9 (line 5-7), it is stated that “Pups were delivered in 4 of 29 male-female pairs. Owing to small N values, observation of delivery was not taken into account for analysis”. This is a weak argument. 4 out of 29 is 14% of your sample. Were females separated from their pups to run the experiment? This would seem to be a serious confound, depending upon exactly when the pups were delivered. Perhaps eliminating these four pairs from the entire experiment would be reasonable?

6. On page 12 (line 16), it is stated that “In the present study, most of the data seemed o not be distributed normally . . .” This is a weak statement. One could run some analyses to determine if the samples are indeed non-normal (Shapiro-Wild statistic).

7. Page 15 (line 9), the phrase “of post hoc tests” can be deleted. More broadly, it would be useful to mention the non-significant p-values parenthetically throughout the Results section, but this decision should be made by the editor.

8. When describing the results in the Discussion, it should consistently be indicated that the significant difference in female vocalizations occurred after a short isolation interval and the effect occurred after a long isolation interval in males. For example, page 16 (line 6) should include “. . . unfamiliar females, after a short isolation interval.” This point should also be reiterated in the Conclusion section.

9. It is stated on page 17 (line 3-4), that “We expected males to show more vocalization to unfamiliar females”. The basis for this expectation should be briefly stated.

10. In general, the Discussion is quite wordy. Attention should be given to avoiding redundancy and eliminating irrelevant details. For example, the description of 30 min vs. 60 min delay could be stated much more concisely on pages 19-20.

11. On page 20, it is suggested that the differences in vocalization by males after 30 min vs. 1 day of isolation may have been due to the sexual refractory period. The typical length of the sexual refractory period in mice should be stated.

12. Page 20 (line 10), “saturated” should be “satiated”.

13. On page 18 (lines 7-9) the issue of differential response to novelty is brought up, but no sources are cited. It would be useful have some examples of how mice respond to novelty.

Reviewer: 2

Comments to the Author(s)

This article has been modified in response to revisions requested on a prior submission to Biology Letters. Having read only this current version of the manuscript, I believe the authors have successfully answered most of the points previously raised. I have a few comments, raised below as I read through the manuscript:

Introduction

In the introduction, there is a bit of back-and-forth between ‘vocal communication’ and ‘autism’, which slightly impairs the flow of ideas. This could be improved by simple text rearrangements.

For instance, the added section (lines 7-11) currently feels out of context because the authors suddenly refer to their work. Using more general terms could be more appropriate. For example, instead of 'In addition, we previously investigated synapses and behaviors in the mutants of Autism susceptibility candidate 2 (AUTS2) gene, and we reported that excitatory synaptic inputs were increased in the forebrain and that social interaction and vocalizations were altered in the mice', one could use 'For instance, mice carrying the Autism susceptibility candidate 2 (AUTS2) gene show an increase in excitatory synaptic inputs in the forebrain, as well as alteration of their social interaction and vocalizations'. This is just a suggestion, which shows how the introduction could be framed better.

- Line 13: 'Playback of male vocalizations in B6 strain'. This strain needs to be defined here as it is where it first appears.

- Currently, the end of the introduction reads 'it is important to measure the sociality of both sexes using the same methods because mouse models and USVs have been recently investigated to clarify the basic mechanisms of developmental disorders [2-4]'. This section needs rewriting, as the suggested connection to developmental disorders is brought about without context.

Methods & Results

The methods and results are well described.

Discussion

- The authors did well in answering various points raised during the past revision rounds, gathered in a new section entitled '4.3 Technical limitations and considerations for experimental design'.

- In this section 4.3, the text reads 'For decades, the sex differences between male-female and female-female USVs have been discussed'. A reference would be useful here, in particular because the following sentence emphasizes the lack of sex difference in this context.

- I would suggest to replace 'mice change their vocal behavior in response to the context of whom they are dealing' by 'mice change their vocal behavior in response to whom they are facing' or similar.

- I would suggest to replace 'if typical social behavior can be observed in the wild-type both as a quality as well as a quantity' by 'if typical social behavior can be described in the wild-type both in quality as well as in quantity' or similar.

- Within section 4.3: I believe that the analysis conducted using only 'recording 1' is valuable (despite, unfortunate as it is, showing no significant findings) and should be included in the main document. This phase indeed represents the 'true' retention effect, in the sense that it comes right after the 30min or 1day isolation period.

- End of section 4.4: 'clarifying roles' should be 'clarifying the roles'

Conclusion

In the last sentence of the conclusion paragraph, I am not sure to understand the use of 'however'. Wouldn't 'in addition' be more appropriate for instance?

===PREPARING YOUR MANUSCRIPT===

===PREPARING YOUR REVISION IN SCHOLARONE===

- If you are providing image files for potential cover images, please upload these at this step, and inform the editorial office you have done so. You must hold the copyright to any image provided.
- A copy of your point-by-point response to referees and Editors. This will expedite the preparation of your proof.

- Ensure that your data access statement meets the requirements at <https://royalsociety.org/journals/authors/author-guidelines/#data>. You should ensure that you cite the dataset in your reference list. If you have deposited data etc in the Dryad repository, please only include the 'For publication' link at this stage. You should remove the 'For review' link.
- If you are requesting an article processing charge waiver, you must select the relevant waiver option (if requesting a discretionary waiver, the form should have been uploaded at Step 3 'File upload' above).
- If you have uploaded ESM files, please ensure you follow the guidance at <https://royalsociety.org/journals/authors/author-guidelines/#supplementary-material> to include a suitable title and informative caption. An example of appropriate titling and captioning may be found at https://figshare.com/articles/Table_S2_from_Is_there_a_trade-off_between_peak_performance_and_performance_breadth_across_temperatures_for_aerobic_scope_in_teleost_fishes_/3843624.

Author's Response to Decision Letter for (RSOS-201529.R0)

See Appendix A.

Decision letter (RSOS-201529.R1)

Dear Dr Kanno,

It is a pleasure to accept your manuscript entitled "Sex differences in vocalizations to familiar or unfamiliar females in mice" in its current form for publication in Royal Society Open Science.

Kind regards,

Andrew Dunn
Royal Society Open Science Editorial Office
Royal Society Open Science
openscience@royalsociety.org

on behalf of Dr Cynthia Downs (Associate Editor) and Kevin Padian (Subject Editor)
openscience@royalsociety.org

Associate Editor Comments to Author (Dr Cynthia Downs):

I reviewed "Sex differences in vocalization to familiar or unfamiliar females in mice." I am satisfied with how the authors have addressed the reviewers' comments. The study described in this manuscript is scientifically sound and the scope of interpretation of the article is appropriate. I have also reviewed the publicly available data on Figshare and see no obvious signs of errors and it appears that the data are sufficient to recreate the analyses.

Appendix A

Dear Dr. Cynthia Downs
Associate Editor

We are grateful to the referees, who reviewed our manuscript for consideration in the *Royal Society Open Science*. Their critical comments and useful suggestions have helped us to improve our manuscript considerably. As indicated in the responses below, we have taken all the comments and suggestions into account in the revised version of our manuscript.

Rephrased parts and additions to our manuscript are written in blue and red text, respectively.

The page and line numbers referring to rephrased parts correspond to the main manuscript (Word file).

Sincerely yours.

Kouta KANNO, Ph.D.

Associate Professor. Lab. of Neuroscience,
Course of Psychology, Department of Humanities
Faculty of Law, Economics, and Humanities
Kagoshima University

E-mail: canno@leh.kagoshima-u.ac.jp

Korimoto 1-21-30, Kagoshima City, Kagoshima,
890-0065 Japan

Tel: +81-99-285-7624

Fax: +81-99-285-7609

Response

Reviewer Comments to Author:

Reviewer: 1

Comments to the Author(s)

This paper describes an experiment designed to test the vocal responses of mice to familiar and unfamiliar females using two different memory retention periods (30 min and one day). Females preferentially vocalized toward unfamiliar females after the 30 min retention period but not after the 1 day period. In contrast, males preferentially vocalized toward familiar females after the 1 day period but not after the 30 min period. The context of this experiment is limited in scope, and the authors are careful not to overstate the results. I reviewed a previously submitted version of this manuscript, and the authors have generally responded to my previous concerns adequately. My remaining concerns are itemized below:

Reply:

We are grateful to Referee 1 for their critical and useful comments. We have taken all comments into account and provide a point-by-point discussion below.

1. The entire manuscript should be edited carefully by a native English speaker. Although many of the grammatical mistakes in the previous draft have been corrected, numerous new mistakes have been added.

Reply:

We apologize for the grammatical mistakes. The present version was edited by Editage (www.editage.com) and we have attached a Certificate of English Editing to this effect.

2. The context of autism treatment is interesting (p. 4), but it would be helpful to explain how USV changed in mice with the manipulation of the AUT2 gene. Briefly mentioning this clinical context in the Discussion would also be useful.

Reply:

As described in the response to comment 4 from Referee 1, we rearranged several sentences. In this part, comments from Referee 2 were taken into account, as well.

3. Page 7 (lines 10-12) should be reworded as “. . . and 24-hour time intervals after female subjects encountered females were used as familiar individuals were applied.”

Reply:

We have corrected the sentence as follows:

P7 line 4–5

In the study by D'Amat and Moles study, the authors applied 15 min, 30 min, 60 min, and 24 h time intervals after female subjects encountered females that were used as familiar individuals..

4. It remains vague regarding exactly what about this experiment was distinct from the results reported by D'Amat and Moles (2001). The main goals of this study, which are not simply a replication of past work, should be more clearly stated at the end of the Introduction.

Reply:

We rearranged sentences to improve their flow. The explanation for the utility of USVs for autism studies was moved and combined with the last paragraph of the Introduction (p. 8). We hope this rearrangement also improves the clarity of our aims in the manuscript.

5. On page 9 (line 5-7), it is stated that “Pups were delivered in 4 of 29 male-female pairs. Owing to small N values, observation of delivery was not taken into account for analysis”. This is a weak argument. 4 out of 29 is 14% of your sample. Were females separated from their pups to run the experiment? This would seem to be a serious confound, depending upon exactly when the pups were delivered. Perhaps eliminating these four pairs from the entire experiment would be reasonable?

Reply:

In general, mice pups are delivered approximately twenty days after fertilization. Recording 1 and 2 were conducted on days 7 or 8 (see, Fig. 1A) which is long before the birth of pups. Therefore, the recordings were not influenced by pup birth.

6. On page 12 (line 16), it is stated that “In the present study, most of the data seemed o not be distributed normally . . .” This is a weak statement. On could run some analyses to determine if the samples are indeed non-normal (Shapiro-Wild statistic).

Reply:

Just because the data are normally distributed, it does not follow that use of nonparametric tests is a statistical violation. Meanwhile, a parametric test should not be used when the data are not normally distributed. In addition, conducting statistical tests several times themselves (including Shapiro–Wilk test or Kolmogorov–Smirnov test) increases the possibility of Type 1 error. Therefore, we often selected nonparametric methods without using such normality tests when the structure of comparison is simple. Conversely, prior normality tests are needed when we have to use parametric methods.

However, the following sentences have been removed because unnecessary sentences are confusing.

“In the present study, most of the data seemed to not be distributed normally; therefore, nonparametric tests were used”

7. Page 15 (line 9), the phrase “of post hoc tests” can be deleted. More broadly, it would be useful to mention the non-significant p-values parenthetically throughout the Results section, but this decision should be made by the editor.

Reply:

We rewrote the section as advised (p. 15).

8. When describing the results in the Discussion, it should consistently be indicated that the significant difference in female vocalizations occurred after a short isolation interval

and the effect occurred after a long isolation interval in males. For example, page 16 (line 6) should include “. . . unfamiliar females, after a short isolation interval.” This point should also be reiterated in the Conclusion section.

&

10. In general, the Discussion is quite wordy. Attention should be given to avoiding redundancy and eliminating irrelevant details. For example, the description of 30 min vs. 60 min delay could be stated much more concisely on pages 19-20.

Reply:

We rephrased several sentences in the discussion to this effect, also taking into account similar comments from Referee 2.

9. It is stated on page 17 (line 3-4), that “We expected males to show more vocalization to unfamiliar females”. The basis for this expectation should be briefly stated.

Reply:

We added a brief explanation and citations on this; p. 18, lines 1–2

We expected males to show more vocalization toward unfamiliar females **because it has been reported that males exhibit more sexual motivation toward unfamiliar novel females (Coolidge effect) [33,34].**

11. On page 20, it is suggested that the differences in vocalization by males after 30 min vs. 1 day of isolation may have been due to the sexual refractory period. The typical length of the sexual refractory period in mice should be stated.

Reply:

We briefly added an explanation of the refractory period length; p. 21, lines 6

This interval between sexual behaviors is called a refractory period and USVs are not observed in this period **(the first refractory period ranges from 30 to 60 min) [37].**

12. Page 20 (line 10), “saturated” should be “satiated”.

Reply:

We replaced the words as recommended (p. 21, lines 8).

13. On page 18 (lines 7-9) the issue of differential response to novelty is brought up, but no sources are cited. It would be useful have some examples of how mice respond to novelty.

Reply:

We added citations for the habituation–dishabituation test (p. 19, lines 9).

Reviewer: 2

Comments to the Author(s)

This article has been modified in response to revisions requested on a prior submission to *Biology Letters*. Having read only this current version of the manuscript, I believe the authors have successfully answered most of the points previously raised. I have a few comments, raised below as I read through the manuscript:

Reply:

We are grateful to Referee 2 for their critical and useful comments. We have taken all comments into account and provide a point-by-point discussion below.

Introduction

In the introduction, there is a bit of back-and-forth between ‘vocal communication’ and ‘autism’, which slightly impairs the flow of ideas. This could be improved by simple text rearrangements. For instance, the added section (lines 7-11) currently feels out of context because the authors suddenly refer to their work. Using more general terms could be more appropriate. For example, instead of ‘In addition, we previously

investigated synapses and behaviors in the mutants of Autism susceptibility candidate 2 (AUTS2) gene, and we reported that excitatory synaptic inputs were increased in the forebrain and that social interaction and vocalizations were altered in the mice’, one could use ‘For instance, mice carrying the Autism susceptibility candidate 2 (AUTS2) gene show an increase in excitatory synaptic inputs in the forebrain, as well as alteration of their social interaction and vocalizations’. This is just a suggestion, which shows how the introduction could be framed better.

&

- Currently, the end of the introduction reads ‘it is important to measure the sociality of both sexes using the same methods because mouse models and USVs have been recently investigated to clarify the basic mechanisms of developmental disorders [2–4]’. This section needs rewriting, as the suggested connection to developmental disorders is brought about without context.

Reply:

We rearranged the sentences to improve the flow. The explanation for the utility of USVs for autism studies was moved and combined with the last paragraph of the Introduction (p. 8).

- Line 13: ‘Playback of male vocalizations in B6 strain’. This strain needs to be defined here as it is where it first appears.

Reply:

We defined the strains as advised (p. 5, line 6).

Methods & Results

The methods and results are well described.

&

Discussion

- The authors did well in answering various points raised during the past revision rounds, gathered in a new section entitled ‘4.3 Technical limitations and considerations for experimental design’.

Reply:

We appreciate this comment from Referee 2.

- In this section 4.3, the text reads ‘For decades, the sex differences between male–female and female–female USVs have been discussed’. A reference would be useful here, in particular because the following sentence emphasizes the lack of sex difference in this context.

Reply:

We added the references as advised (p. 19, line 1).

- I would suggest to replace ‘mice change their vocal behavior in response to the context of whom they are dealing’ by ‘mice change their vocal behavior in response to whom they are facing’ or similar.

Reply:

We rephrased the sentence as advised (p. 19, lines 4–5).

- I would suggest to replace ‘if typical social behavior can be observed in the wild-type both as a quality as well as a quantity’ by ‘if typical social behavior can be described in the wild-type both in quality as well as in quantity’ or similar.

Reply:

We rephrased the sentence as advised (p. 19, lines 14–15).

- Within section 4.3: I believe that the analysis conducted using only ‘recording 1’ is valuable (despite, unfortunate as it is, showing no significant findings) and should be included in the main document. This phase indeed represents the ‘true’ retention effect, in the sense that it comes right after the 30min or 1day isolation period.

Reply:

We added a sentence referring to this in the Results (3.1 and 3.2), and also rephrased the Discussion to capture this (p. 20, lines 7–9).

- End of section 4.4: ‘clarifying roles’ should be ‘clarifying the roles’

Reply:

We corrected this as advised (p. 24, line 1).

Conclusion

In the last sentence of the conclusion paragraph, I am not sure to understand the use of ‘however’. Wouldn’t ‘in addition’ be more appropriate for instance?

Reply:

We corrected this as advised (p. 24, line 13).